# *Clostridioides difficile* Infection, Still a Long Way to Go

**DOI:** 10.3390/jcm10030389

**Published:** 2021-01-20

**Authors:** Eleftheria Kampouri, Antony Croxatto, Guy Prod’hom, Benoit Guery

**Affiliations:** 1Infectious Diseases Service, Department of Medicine, University Hospital and University of Lausanne, 1011 Lausanne, Switzerland; eleftheria-evdokia.kampouri@chuv.ch; 2Institute of Microbiology, Department of Medical Laboratory and Pathology, University Hospital and University of Lausanne, 1011 Lausanne, Switzerland; antony.croxatto@chuv.ch (A.C.); guy.prodhom@chuv.ch (G.P.)

**Keywords:** clostridium difficile, *Clostridioides difficile*, epidemiology, diagnosis, treatment, severity, carriage, prevention

## Abstract

*Clostridioides difficile* is an increasingly common pathogen both within and outside the hospital and is responsible for a large clinical spectrum from asymptomatic carriage to complicated infection associated with a high mortality. While diagnostic methods have considerably progressed over the years, the optimal diagnostic algorithm is still debated and there is no single diagnostic test that can be used as a standalone test. More importantly, the heterogeneity in diagnostic practices between centers along with the lack of robust surveillance systems in all countries and an important degree of underdiagnosis due to lack of clinical suspicion in the community, hinder a more accurate evaluation of the burden of disease. Our improved understanding of the physiopathology of CDI has allowed some significant progress in the treatment of CDI, including a broader use of fidaxomicine, the use of fecal microbiota transplantation for multiples recurrences and newer approaches including antibodies, vaccines and new molecules, already developed or in the pipeline. However, the management of CDI recurrences and severe infections remain challenging and the main question remains: how to best target these often expensive treatments to the right population. In this review we discuss current diagnostic approaches, treatment and potential prevention strategies, with a special focus on recent advances in the field as well as areas of uncertainty and unmet needs and how to address them.

## 1. Introduction

*Clostridioides* (formerly *Clostridium*) *difficile* is a major cause of healthcare-associated diarrhea, and is increasingly present in the community. A lot has changed in our understanding of the physiopathology of this frequent pathogen as well as in the diagnosis and treatment of *Clostridioides difficile* infection (CDI). However, several questions remain unanswered. Diagnostic approaches and surveillance systems vary considerably between regions hindering an accurate estimation of the global burden of CDI. Furthermore, diagnosis remains suboptimal, especially in certain settings, such as the community. The management of recurrences, severe and complicated disease, and the optimal use of new therapeutic molecules to target the right population remain challenging. Finally, areas of uncertainty persist regarding the significance of asymptomatic carriage, the optimal clinical endpoints and the long-term follow-up and outcomes of these patients. We selected a number of publications by searching into Pubmed to review the epidemiology, clinical presentation, outcomes and management of CDI. We discuss current diagnostic approaches and treatment options, with a special focus on areas of uncertainty and recent advances in the field.

## 2. Epidemiology

The incidence of *Clostridioides difficile* infection markedly increased worldwide in the 2000s [1,2,3], in part due to the emergence and rapid spread throughout North America and Europe of the virulent, epidemic ribotype 027 strain (North-American Pulsefield type 1, NAP1/027), which was associated with increased severity of disease and mortality [4,5,6]. At the same time, the introduction of more sensitive diagnostic assays, such as nucleic acid amplification assays (NAAT), seems to have contributed to a substantial increase in the reported CDI incidence [7,8]. Further adding to an already high burden, recurrences after diagnosis of CDI are frequent, with 10–30% of patients developing at least one recurrence and the risk increasing with each successive episode [9,10]. To be able to accurately evaluate the burden of CDI, there is a need for standardization of diagnostic algorithms and a robust surveillance system, and this need is not entirely nor universally met.

In the US, an estimated 453,000 cases of *C. difficile* infection occurred in 2011 based on data from active population- and laboratory-based surveillance across different geographic areas, resulting in approximately 29,000 deaths [3]. On a more positive note, according to a more recent study by Guh et al., the estimated burden of CDI decreased in the US between 2011 and 2017, probably as a result of improved infection control measures and a concomitant overall decline of healthcare-associated infections (HAIs) [11].

Epidemiological data are scarcer in Europe. The lack of a standardization of diagnostic procedures in hospitals, as well as the heterogeneity in the presence and the methodology of national surveillance and the availability of molecular typing, hinder a more accurate overview of the burden of CDI. In a study conducted in 34 European countries in 2008, the incidence and distribution of causative ribotypes varied greatly between countries, with an overall incidence of 4.1 per 10,000 patient-days per hospital [1]. *C. difficile* was the sixth most frequent pathogen responsible for healthcare-associated infections in a European point prevalence study conducted in 2016–2017, with an annual estimated number of cases of 189,256 [12].

Continued molecular typing is also important for a better understanding of the current epidemiology as well as in order to timely detect emerging *C. difficile* strains. For instance, even though the prevalence of 027 ribotype is decreasing in Europe [1,13], the emergence of a virulent strain ribotype 078 has been reported in the Netherlands, with an increasing prevalence between 2005 and 2008 and a severity similar to that reported with ribotype 027 [14]. The unmet needs in epidemiology are summarized in Box 1.

Box 1Unmet Needs in Epidemiology.Despite the important burden of CDI, diagnostic methods and surveillance vary across regions and countries in Europe, hampering a global and more precise overview of the burden of CDI. There is an urgent need to obtain European data on CDI burden in hospitals and in the community.

## 3. Diagnosis and Microbiology

Despite the great progress in diagnostic methods and the availability of international guidelines [15,16], the ideal diagnostic algorithm is still a matter of debate. The large clinical spectrum from asymptomatic carriage of a toxigenic strain to complicated disease and variability of outcomes, highly contribute to the confusion.

During the last decades, major changes occurred in diagnostic methods for the detection of toxigenic CDI. Previously, the main laboratory methods included toxigenic culture (TC) and cell cytotoxicity assays (CCA) and while these methods are still considered to be the reference methods for the diagnosis of CDI, they are no longer routinely used in conventional diagnostic laboratories. These methods present several limitations including the slow time to results (turnaround time of two to four days) and require high workloads performed by expert microbiologists rendering them not suitable to process large sample volumes with a high throughput [17]. These reference methods have been thus replaced by easier-to-use, rapid tests (around 30 min to 4 h) with little hands-on time such as direct toxin A/B enzyme immunoassay (EIA) EIA, lateral flow immunoassays (LFA) and NAAT [18].

EIA and rapid tests such as LFA (15–30 min) present high sensitivity and specificity for the detection of CD antigen glutamate dehydrogenase (GDH). However, a high specificity (97–100%) but a low sensitivity (29–86%) is observed for toxin A/B EIA or LFA detection, depending on the test and the patient population tested [18]. This low sensitivity of LFA for toxin detection excludes its use as a standalone approach and underlines the need for additional different tests to exclude the presence of toxigenic CD with a high negative predictive value (NPV). Before the introduction of NAAT in routine diagnostic facilities, GDH positive and toxin A/B negative EIA/LFA tests required additional analysis by usually performing selective culture methods for CD isolation and enrichment to repeat toxin A/B LFA or EIA assays to achieve higher sensitivity. Nowadays, this culture approach, requiring several days, has been replaced mostly by rapid NAAT assays for the detection of toxigenic *C. difficile* strains, following GDH positive EIA/LFA assays.

Most NAAT assays detect only the toxin A/B encoding genes *tcdA* and/or *tcdB*, which are usually sufficient for the diagnosis. Some NAAT assays (eg., Verigene, Xpert *C.difficile*) include additional important clinical and epidemiological gene targets by combining the detection of the toxin A/B encoding genes *tcdA*/*tcdB* with the detection of the binary toxin genes (*cdt*) and a deletion at nucleotide position 117 on the regulatory *tcdC* gene present in CD ribotype 27 strains and other related isolates [17]. The combined detection of the toxin B and the binary toxin is a prognostic factor of severe CDI [19]. The detection of CD ribotype 027 is important since the mutation at nucleotide position 117 of the regulatory *tcdC* gene can be associated with an increased toxin production (hypervirulence) and enhanced spore formation. Enhanced spore formation is associated with increased environmental and healthcare persistence favoring the emergence of epidemiological outbreaks [19]. Thus, NAAT assays offer the possibility of rapid detection of hypervirulent CDI ribotype 027, allowing a more stringent healthcare surveillance system. The rapid CDI diagnosis provided by EIA/LFA and NAAT compared to slow time to results of TC and CCA has significantly improved infection control management to prevent CDI transmission in healthcare facilities. Moreover, the use of NAAT allows rapid detection of CDI from symptomatic patients and with a high sensitivity that is essential to take rapid preventive measures.

### 3.1. Why Are We Not Using NAAT as the Ultimate Tool for CDI Diagnosis?

Current commercialized NAAT assays used in routine diagnostic laboratories are only qualitative (positive or negative) and cannot characterize the bacterial load and the viability of CD (viable or dead bacteria) in stool samples. NAAT, as a standalone test, is not appropriate to provide an adequate clinical positive predictive value (PPV) with low CDI prevalence [18]. Due to their high sensitivity, positive NAAT assays require a thorough and sometimes difficult clinical evaluation to discriminate CDI from (1) asymptomatic carriage of live toxigenic CD; (2) DNA from dead bacteria; and (3) long-lasting bacterial shedding following treated CDI [18,20]. However, these highly sensitive assays do not usually require further testing to exclude CDI with a high negative predictive value (NPV).

The recommended two- to three-step algorithms combining EIA/LFA, NAAT and TC proposed by the European Society of Clinical Microbiology and Infectious Diseases (ESCMID) guideline certainly improved PPV but are still insufficient to cover, with high PPV and NPV, the complex spectrum of CDI clinical presentations and transmission [18]. The cornerstone of the proposed two- to three-step algorithms is the detection of free toxins by EIA/LFA techniques, either directly from stools or following TC. However, although presenting high clinical specificity, EIA/LFA present reduced sensitivity compared to NAAT, implying that negative EIA/LFA results with positive NAAT assays cannot exclude with a high NPV active disease but with low toxin concentrations. Clinical evaluation of all cases presenting with NAAT positive and toxin A/B EIA (NAAT+/EIA-) negative results showed that 46.4% of the patients were colonized and 53.6% presented active disease [21]. Polage et al. reported that among NAAT+/EIA-, 38% were toxin positive by the reference method CCA [22]. However, many studies clearly demonstrate that NAAT+/EIA+ results are usually associated with higher CD bacterial loads, more severe symptoms and higher mortality rate than NAAT+/EIA- [22,23,24,25].

### 3.2. Could Quantitative NAAT Be Used to Predict Clinical Outcome?

Since a toxin A/B EIA/LFA positive assay is associated with higher bacterial loads, quantitative NAAT could have the potential to provide toxin quantification corresponding to positive toxin EIA/LFA and CCA, that could be used as a marker of infection severity and clinical outcomes. Davies et al. investigated the potential utility of toxin gene quantitative NAAT by determining the predictive value of low cycle threshold (Ct) for toxin positivity, CDI severity, mortality and CDI recurrence [26]. Unfortunately, the authors only observed a limited specificity and sensitivity of quantitative NAAT for these clinical parameters excluding its use as a standalone test. These preliminary study results demonstrate that accurate CDI diagnosis will likely require the use of combined direct (toxin detection and quantification) and indirect (infection and inflammation markers) assays to obtain optimal PPV and NPV. Moreover, the shedding of CD in stools is not constant and a variation in bacterial/toxins loads in stools collected from the same patients during a short period of time has been observed, limiting the utility of quantitative approaches as a prognostic marker.

### 3.3. Why Do We Not Routinely Perform Antibiotic Susceptibility Testing (AST) for CDI?

Antibiotic susceptibility testing (AST) is not routinely performed for CDI due to the absence of interpretation criteria from EUCAST and/or method uncertainty for currently used treatments, namely metronidazole, vancomycin and fidaxomicin. For vancomycin and metronidazole, the breakpoints are based on epidemiological cut-off values (ECOFFs) and applied to oral treatment. As stated by EUCAST, there is no conclusive data regarding the relation between MICs and outcome for these two antibiotics. For fidaxomicin, no breakpoints and ECOFF have been set by EUCAST since major variations in MIC distribution between studies have been observed (The European Committee on Antimicrobial Susceptibility Testing. Breakpoint tables for interpretation of MICs and zone diameters. Version 10.0, 2020).

### 3.4. Are We over- or Underdiagnosing CDI?

Fully comprehending the burden of disease requires a profound understanding of the diagnostic methods used and their respective weaknesses and merits. A growing body of evidence suggests that the detection of free toxin in stools by EIAs correlates with clinical symptoms and outcomes while the mere presence of a positive NAAT could lead to overdiagnosis and an overestimation of the real incidence of CDI [22,24,25,27]. On the other hand, despite the higher specificity of EIAs for toxins, detection of free toxins may lack sensitivity and EIAs can be negative in the early stages (due to the smaller bacterial burden) and in patients with complicated disease [28].

Several studies suggest a considerable degree of underdiagnosis in Europe. In a prospective point prevalence study conducted in 482 European centers on two sampling days, 23% of all positive CDI samples, as determined by the reference national laboratory, were not diagnosed by participating hospitals due to a lack of clinical suspicion and suboptimal laboratory methods. As a result, an estimated 40,000 inpatients are potentially undiagnosed per year [29]. In another Spanish study where 807 specimens were addressed to a reference laboratory from 118 participating centers, two out of every three episodes of CDI were not diagnosed due to lack of clinical suspicion or non-sensitive techniques [30].

This “underdiagnosis” seems to be even more relevant in the community, where the main problem is the lack of clinical suspicion and limited awareness among physicians. While CDI is identified as the leading cause of hospital-acquired diarrhea and easily suspected in this setting, its role in the community is less clear. More than a quarter of all CDI cases are attributed to community acquisition and occur frequently in the absence of traditional risk factors, such as advanced age or prior antibiotic exposure [3,31,32]. In a large US study, an estimated of 345,000 cases occurred outside of the hospital with almost half of them considered purely community-associated (CA) (and so by definition with no healthcare exposure in the previous 12 weeks) [3]. However, another study using the same surveillance program but data from earlier years showed that some kind of exposure, such as visit in an outpatient healthcare setting, was present in 80% of CA cases [33]. More studies from Europe further highlight the role of *C. difficile* as a community pathogen. In a population-based cohort a significant proportion of CDI cases were CA (41% of 385) and occurred in younger patients who presented less severe disease [31]. In a nationwide population-based study in Finland, one third of all CDIs were CA and again patients were younger and mortality was lower in comparison to hospital-acquired CDI (3.2% vs 13.3%, *p* < 0.001) [32].

### 3.5. So Where Exactly Is the Problem?

According to one Dutch study, when CDI testing was performed in all unformed stool samples submitted by general practitioners (GPs) in search for any enteric pathogen, 1.5% were positive for *C. difficile* (out of 12,714 samples). This rate was comparable to other classically community-acquired pathogens, such as *Salmonella* spp. Interestingly, CDI testing was requested by GPs for only 7% of all samples, which would lead to potential missed diagnosis in 60% of all CDIs [34]. This rate of positive CDI testing is similar to other studies performed in a comparable setting [35,36]. Similarly, when stool samples were tested irrespectively of GPs’ request in 15 different laboratories in France, the incidence of toxigenic *C. difficile* as detected by toxigenic culture was 3.27% and 1.81% by a positive cytotoxicity assay. In this study, *C. difficile* was the second more frequent pathogen after *Campylobacter* spp. GPs requested *C. difficile* testing in only 13% of all stool samples thereby detecting only half of all potential CDI cases. It is worth mentioning that among patients with positive CDI testing, more than half had not been hospitalized within 12 weeks (CA-CDI) [37]. These data highlight an urgent need to raise awareness among physicians regarding CDI in the community, which can present with a less severe disease and affect younger patients, with a lower comorbidity load and in the total absence of healthcare exposure. The unmet needs in diagnosis are summarized in Box 2.

Box 2Unmet Needs in Diagnosis.Accurate CDI diagnosis cannot be achieved with a single assay.NAAT, as a standalone test, is not appropriate to provide an adequate clinical PPV with low CDI prevalence, although some controversy remains.The recommended two- to three-step algorithms combining EIA/LFA, NAAT and TC proposed by the ESCMID guideline is a diagnostic improvement but is insufficient to cover with high PPV and NPV the complex spectrum of CDI clinical presentations and transmission. Nevertheless, EIA/LFA toxin assays should be avoided due to relatively low sensitivity.Even though the role of *C. difficile* is increasingly being recognized in the community, this diagnosis is not systematically suspected in the community setting and in the absence of traditional risk factors.More population-based studies are required to better appreciate the true burden of disease in the community.There is a need to provide specific recommendations for testing for *C. difficile* in patients with diarrhea outside the hospital and an increase in physicians’ awareness.

## 4. Definitions and Endpoints

An important concern in most studies evaluating treatment strategies is a considerable amount of heterogeneity in definitions used. From the definition of severe infection to what constitutes a “clinical cure” and a “sustained clinical cure” and when to assess these endpoints, the lack of harmonization hinders comparisons between studies and, in some cases, a better understanding of the impact of treatments. There is a need for strong and uniform definitions as well as the development of new and more objective tools to assess the response to treatment.

### 4.1. What Does Severe Infection Mean?

A lot of confusion exists around the definition of severe infection, which is sometimes used interchangeably with fulminant or complicated infection [38]. In the more recent IDSA/SHEA guidelines, a fulminant or complicated episode refers to infection complicated by hypotension, shock, ileus or megacolon [16]. In this setting, severe disease englobes the presence of prognostic factors associated with unfavorable outcomes, namely complicated disease, treatment failure, intenseive care unit (ICU) admission or mortality.

Several studies evaluate potential factors associated with disease severity and treatment outcome. In a large database of two randomized controlled trials (1105 patients) [39,40], Bauer et al. showed an association with treatment failure of three quantitative parameters: fever (RR 2.45; 95% CI 1.07–5.61), leukocytosis (>10,000 cells/mL) (RR 2.29; 95% CI 1.63–3.21) and renal failure (RR 2.52; 95% CI 1.82–3.50) [41]. Notably, different timing of measurement of leucocyte count and serum creatinine level led to important variation in severity classification, underlining the need for a strict definition of the timing of measurement. Miller et al. analyzed the database of the same two clinical trials to develop a clinical tool for severity stratification based on a combination of five clinical and laboratory variables: age, antibiotic treatment, leucocytes, albumin and serum creatinine. The derived score (ATLAS) seemed to perform well in predicting response to treatment, though optimal cutoff is not clear [42]. Lungulescu et al., in a retrospective cohort study of 255 patients, identified four risk factors associated with severe disease in univariate analysis (history of malignancy, white blood cell count of more than 20 G/l, hypoalbuminemia and a rise in creatinine) and developed a CDI severity index score based on these variables upon admission. A cut-off value of two had a sensitivity of 82% and specificity of 65% in predicting severe disease, defined here as need for colectomy, ICU admission, hospitalization >10 days or death [43]. Finally, a systematic review of studies evaluating clinical prediction rules for unfavorable outcomes showed that, with the exception of leukocytosis, albumin and age, there was significant heterogeneity in the variables used by most studies. The authors conclude that important methodological limitations of many of these studies and the small sample sizes limited the use of the prediction rules in clinical practice [44].

Adding to the existing confusion, the prognostic factors and thus the definition of severe infection differs between ESCMID and IDSA/SHEA guidelines (Table 1). In the current ESCMID guidelines, a wide variety of prognostic factors (patient’s characteristics, clinical features, laboratory findings, imaging and endoscopy) are proposed to help distinguish patients at increased risk for severe disease and determine treatment choices. Among these, four factors are classified with a strong recommendation, namely: age > 65 years old, leucocytosis, decreased albumin and rise in serum creatinine (38). For practical reasons, the criteria proposed by the current IDSA/SHEA guidelines continue to be marked leucocytosis (>15,000 cell/mL) or a serum creatinine level> 1.5 mg/dL (133 μmol/L) [16]. The need for validation of these criteria in large cohorts is highlighted in this guideline, as is the suboptimal performance in certain patient groups such as in patients with hematologic malignancies [45] or renal insufficiency [46], which could lead to a suboptimal treatment choice. Importantly, in the group of patients with hematological malignancies, a case-control study matching 41 inpatients with hematologic malignancies to 82 control patients showed that creatinine levels and WBC counts tended to be lower in the hematologic patients and may not be applicable in this group [45]. There is an urgent need to develop and validate better tools to estimate disease severity in special groups.

### 4.2. What Is an Adequate Endpoint?

Although highly variable between studies, one frequently used clinical endpoint is a clinical cure. Clinical cure is usually evaluated according to the “investigator’s assessment”, hence it can be subject to important bias. Another important point remains the time of evaluation of sustained clinical cure. In fact, in the absence of a strict recommendation regarding what constitutes a sustained clinical cure, a time interval between 30 and 90 days [39,40,47,48] is often used. Using a 30-day time window to evaluate a sustained cure would probably miss the proportion of recurrences occurring after the first month and until 8 weeks (late recurrences). It appears logical that a more generous 90-days time limit would be more appropriate to better evaluate the therapeutic impact of different strategies and, evidently, endpoints should be harmonized among studies to allow comparability.

## 5. Treatment

### 5.1. Antibiotic Treatment

#### 5.1.1. Mild/Moderate and Severe Forms

Three drugs are still currently used in the treatment of CDI, metronidazole, vancomycin, and fidaxomicin. Depending on the clinical form, the risk factors potentially associated to recurrences and the possibility to take an oral treatment, these drugs are prescribed according to international guidelines from the ESCMID or the SHEA/IDSA [16,38]. A clinical cure per se is not a real challenge and the results are acceptable. Focusing on the first two drugs in randomized clinical trials, metronidazole and vancomycin, a clinical cure is obtained globally in more than 80% of the cases (Table 1). The main difference is observed in severe forms with rates of clinical cure as low as 66% with metronidazole [49]. Again, regarding clinical cures, no differences are observed between vancomycin and fidaxomicin as shown in Table 2. As recently suggested by the IDSA/SHEA, only vancomycin and fidaxomicin should be used to treat non-severe as well as severe forms of CDI [16]. Another potential question when using fidaxomicin is whether the treatment should be administered for the traditionally recommended duration of 10 days [39,40] or as an extended administration (treatment administered over 25 days instead of 10) [48]. In the initial paper suggesting better results with an extended administration of fidaxomicin over 25 days, the control group was vancomycin for 10 days [48]. In this study, patients ≥ 60 years presented high risk of recurrence. Interestingly, recurrences at day 90 reached only 19% in the vancomycin group, which is lower than the rate reported in any of the previous trials comparing these two drugs. To have a definite answer, it would be interesting to compare fidaxomicin for 10 days, versus extended-pulsed administration of fidaxomicin as well as vancomycin.

#### 5.1.2. Complicated Forms

Complicated forms are fortunately less frequent and it is therefore difficult to obtain high quality randomized studies. Vancomycin is administered orally, generally using a high dosage even though the rationale is not very strong [16]. In case of ileus, vancomycin should be administered locally per rectum [53,54]. In both cases, intravenous metronidazole is generally associated [55]. Other treatments such as tigecyclin or intravenous immunoglobulin have been proposed in fulminant CDI, but their efficacy has not been evaluated in randomized trials [56,57,58,59].

#### 5.1.3. Recurrences

It seems important to differentiate the first recurrence from the following episodes. While metronidazole is clearly not indicated, vancomycin and fidaxomicin remain potential options in this indication [16]. In a first recurrence, data from two phase 3 randomized trials comparing vancomycin to fidaxomicin were analyzed, showing that second recurrence was less frequent following fidaxomicin treatment [60]. This advantage of fidaxomicin is not confirmed upon further episodes [61]. Tapered and pulsed-dosed vancomycin are currently used for second and subsequent episodes [62] and Fecal Microbiota Transplantation (FMT) is proposed after the second recurrence based on the trial published by Van Nood et al. in 2013 [63]. It could be relevant to study the potential effect on recurrence when FMT is proposed at the very first recurrence. Such a trial could compare several strategies differentiating between the first and further recurrences. Regarding the first recurrence, it would be important to compare vancomycin pulsed or tapered, to extended pulsed fidaxomicin and FMT. For further recurrences, it has been previously shown that fidaxomicin was not efficient, so evaluation should be limited to only pulsed vancomycin and FMT.

### 5.2. Surgery

Colectomy as a treatment in the setting of fulminant CDI has been extensively discussed in the literature. In a population-based study with 67 patients who required colectomy, mortality reached 48% [64], underlining the severity despite surgical intervention. In a retrospective study including 165 cases of CDI admitted in the ICU, 53% cases [65] died, but emergency colectomy was associated with better prognosis [66]. Another study showed comparable results with a 30-day mortality measured at 57% (28/49) and a 5-year survival at 38% (8/21), in 16.3% for all patients [67]. Interestingly, a conservative surgical approach associating loop ileostomy with anterograde vancomycin lavage [68] was associated with a major reduction in mortality compared to historic controls from 50 to 19% (*p* < 0.006). A more recent retrospective study included 3201 patients, among whom 613 underwent loop ileostomy, and 2408 subtotal colectomies [69], and showed no difference in in-hospital mortality (25.96% vs 31.18% respectively, *p* = 0.28). The analysis of these studies presenting major limitations regarding their design or the number of patients involved, suggesting that a more conservative approach could be preferable. However, the selection of patients requiring these surgeries needs to be clarified and the criteria required to choose between loop ileostomy versus colectomy are still not precisely described in the current guidelines [16,38].

### 5.3. Monoclonal Antibodies

In 2010, a randomized double placebo-controlled study showed that two neutralizing monoclonal antibodies directed against toxin A and B of *C. difficile* significantly reduced the recurrences (from 25% to 7%) after a treatment with metronidazole or vancomycin [70]. Two randomized double placebo-controlled phase 3 trials, MODIFY I and II, confirmed these results and showed that bezlotoxumab (antibody directed against the toxin B) was responsible for the effect and associated to a lower rate of recurrent infection than placebo (from 27% to 17%) [71]. In the recent IDSA/SHEA guidelines, there is no current recommendation for the prescription of bezlotoxumab [16]. One of the main problems is the definition of the target population and the cost-efficiency relationship. Regarding the efficacy, the rate of recurrence after bezlotoxumab is comparable to the rate observed with the classic administration of fidaxomicin, both compared to vancomycin (from 25% to almost 15%). It has been shown that fidaxomicin was cost-effective in patients with severe CDI and patients with a first recurrence in comparison to vancomycin [72]. Comparable data were obtained with bezlotoxumab compared to placebo, where the molecule was proven to be cost effective in the subgroups of patients aged >65 years old, immunocompromised, and with severe CDI [73]. Using the same Markov model, another study showed cost-effectiveness of bezlotoxumab compared to standard of care in 5 subgroups: >65 years old, severe CDI, immunocompromised, >1 CDI episode in the previous 6 months, and >65 years old and with a >1 CDI episode in the previous 6 months [74]. It would be interesting to build a study comparing a standard of care with fidaxomicin usual or extended pulsed administration in these subgroups of patients. Targeting the correct population was also evaluated using prespecified factors with a post hoc analysis of the MODIFY trials [75]. The factors were age >65 years old, history of CDI, compromised immunity, severe CDI, and ribotype 027/078/244. The results confirmed the reduction of recurrences mostly in the group with >3 risk factors. These studies have identified some target groups to propose bezlotoxumab but well-designed randomized studies comparing bezlotoxumab to fidaxomicin in these high-risk patients are missing and definitely needed. A last potential indication not yet completely evaluated is highlighted by a Finnish real-world study, where 8 patients waiting for FMT received bezlotoxumab and remained free of recurrence without FMT [76].

### 5.4. Fecal Microbiota Transplantation (FMT)

FMT was recently proposed in the international guidelines [16,38] in recurrent CDI following the publication of clinical trials showing the superiority of this procedure compared to classic antibiotic treatment [63,77]. From the published studies on FMT, it is not yet completely clear whether an upper or lower route of administration is more effective. A systematic search of 14 studies with data from 305 patients suggested that the lower route was superior [78] but there are no randomized studies providing strong data on this issue. Another parameter to take into account is the number of procedures required depending on the route of administration. In the initial trial with upper administration, the first infusion was associated with an 81.3% cure without relapse and further infusions increased this percentage to 93.8% [63]. Using the lower route, another study showed that the first infusion was associated to a 65% cure and reached 90% with multiple infusions (from one to four additional procedures) [79]. Building a trial comparing the different routes of administration would need to take into consideration the potential efficacy per procedure.

To make the procedure easier, specifically regarding the constraints on the donor side, several studies have already shown that frozen material was comparable to fresh stools [79,80,81,82]. The next question in FMT is the use of new galenics to deliver the transplant. Oral capsules have been shown non-inferior to colonoscopy [83]. Lyophilization was also shown to be efficient [84] but was inferior to fresh or frozen materials [85]. Independently of the galenic used, the main remaining problem is the potential safety of the administered material to the patient. The selection of the donor is very specific and complicated with significant variations among countries [86]. In this setting, the idea would be to obtain an artificial product, not donor related, which raises the question of what is associated to the clinical success in the product administered. As previously discussed, the rationale behind FMT is to correct CDI-associated dysbiosis, especially in recurrent episodes [63,65]. In this setting, microbial diversity of the colonic microbiome could be a potential endpoint per se, but is definitely not the only parameter. Function associated with bacteria may also be important as underlined in a very nice study where, in some patients, cure was obtained independently of diversity restoration [87]. To go a little further, a preliminary investigation showed in 5 patients that sterile filtrate from donor stool was sufficient to restore normal stool habits and eliminate symptoms [88]. The authors suggest a potential role of bacterial cell wall components or DNA fragments, but bacteriophage could also be involved. These results questioned the rate of viability of the microflora of FMT stools preparation (fresh and frozen). Based on these data, one of the most important gaps in knowledge in CDI and FMT is a better understanding of the mechanism associated with success. Identifying these factors could potentially favor more targeted approaches such as the manipulation of the gut microbiota to prevent recurrences in high-risk patients.

### 5.5. Why Is Long-Term Follow up after FMT Recommended?

A critical question regarding FMT are the long-term consequences. There are currently growing evidences in the literature that gut microbiota is involved in a lot of different areas including metabolism, immunology, the lung gut axis, the brain gut axis and the response to cancer therapy [89]. Administration of “healthy” donor microbiota in patients with recurrent CDI may cure the disease, but data from long-term follow up of both donors and recipients are needed to fully understand the impact of this intervention. Even though some data are available from selected, small populations [90,91], there is an imperative need for long term follow-up implementation and wider well-designed studies. The unmet needs in treatment are summarized in Box 3.

Box 3Unsolved Questions and Unmet Needs in Definitions and Treatment.What is the definition of severe infection?What is the optimal clinical endpoint for studies?There is a need for robust definitions and harmonization of endpoints as well as tools for a more objective evaluation of these endpoints.In complicated forms, what is the best antibiotic treatment?Is extended-pulsed fidaxomicin superior to classic 10 days fidaxomicin administration or pulsed vancomycin?What are the selection criteria to choose between loop ileostomy versus total or subtotal colectomy?Can FMT be proposed earlier, after the first recurrence?In a high recurrence risk patient is it more efficient and/or cost-efficient to propose bezlotoxumab or fidaxomicin?Could bezlotoxumab be proposed to avoid FMT in specific populations or where FMT is not available?In FMT, what is the best way to deliver the transplant, upper or lower?Determine the mechanism associated with the efficacy of FMT is crucial to unveil other potential options for treatment.Long-term follow-up of both donors and recipients after FMT with registries at a national level is required.

## 6. Prevention

### 6.1. Prevention and Antibiotic Exposure

Another area where strong data are missing is the prevention of CDI in patients with a previous history of CDI and requiring a new systemic antibiotic treatment. Extension of CDI treatment to cover the time of systemic antibiotic does not seem efficient. If the antibiotic treatment is administered after the initial CDI episode end of treatment, two retrospective cohorts showed a decreased recurrence risk with a various regimen of vancomycin [47,92], but we do not have prospective randomized study to confirm these preliminary data.

### 6.2. Asymptomatic Carriage and Potential Dissemination

*C. difficile* colonization is more frequent than active disease in the hospital, with a prevalence estimated at 3–26% among adult patients in acute care hospitals [93,94]. Previous studies have suggested that asymptomatic carriers of toxigenic strains could be a potential reservoir for hospital-acquired infections via direct patient-to-patient transmission and hospital environment contamination [95,96,97,98,99,100]. However, the role of asymptomatic carriage as a source of spread in the hospital is controversial and the impact on subsequent development of active disease is not yet fully understood. Therefore, routine surveillance of *C. difficile* colonization is not recommended by international guidelines from the ESCMID or the SHEA/IDSA [16,38].

Earlier studies suggest that long-standing asymptomatic colonization could have a protective effect against developing active CDI, partially mediated by humoral immunity against *C. difficile* toxins A and B [101,102,103]. Another potential mechanism would be a protective effect against colonization by toxigenic strains through competition for nutrients and access to the mucosa as it was performed with non-toxigenic strains of *C difficile* [104]. However, other studies suggest an increased risk of CDI among asymptomatic carriers of toxigenic strains. It is plausible that risk of progression from carriage to infection dynamically decreases over time. In a meta-analysis of nineteen studies by Zacharioudakis et al., where the pooled colonization rate upon hospital admission was 8.1%, preceding colonization was associated with a 6-fold increase of the risk of subsequent CDI (RR 5.86; 95% CI 4.21–8.16). Hospitalization history within 12 weeks was the main risk factor for colonization as opposed to previous antibiotic use and history of CDI, which did not significantly impact the risk of colonization [105].

### 6.3. Could the Identification of Carriers Be an Effective Preventive Strategy for HA-CDI?

Curry et al., using a molecular typing technique (Multilocus variable number of tandem repeats analysis, MLVA) showed that a quarter of all isolates from healthcare associated CDI were highly related to isolates from asymptomatic patients identified upon admission screening (isolates recovered from perirectal swabs collected for Vancomycine Resistant Enterocccus surveillance) [93].

One controlled quasi-experimental study in Canada compared the efficacy of universal screening for *C. difficile* carriage and patient isolation versus no screening in the decrease of CDI incidence. Among approximately 7600 patients, 4.8% were identified as potential carriers and 38 patients developed an HA-CDI resulting in an incidence of 3 per 10,000 patient-days, as opposed to 6.9 per 10,000 patient-days in the pre-intervention control period (*p* < 0.001). The authors conclude that detecting and isolating carriers could lead to a significant decrease in incidence of HA-CDI and could be a promising preventive strategy [106]. Of course, another key question regarding *C. difficile* carriage screening strategies would be who to test, the two possible strategies being: universal screening of every patient admitted or target screening of high-risk patients (such as immunocompromised patients and those requiring long antibiotic treatment, etc.) To answer this question there is a need for more large-scale studies with longer follow-ups. The unmet needs in prevention are summarized in Box 4.

Box 4Unsolved Questions and Unmet Needs for Prevention.What is the best preventive regimen to prevent CDI recurrence in patients with an initial episode exposed to systemic antibiotics?The significance of asymptomatic carrier status for the individual patient and their environment in the short term and in the long term as well as their role in hospital transmission is not yet elucidated.There seems to be a potential role of screening for colonization in selected settings and patients upon admission to prevent HA-CDI but more studies are required to better support this strategy.

## 7. Conclusions

After several decades in which *C difficile* infection was of little interest, both diagnostically and therapeutically, recent years have seen the emergence of new diagnostic approaches and new molecules, all of which are based on increasingly solid pathophysiological bases. The initial problem was confined to an infection by a pathogen with the only proposal being eradication by the use of antibiotics, now we are finally measuring the intimate relationships between *C difficile*, the rest of the microbiota and the immune response, the host status. We are only at the beginning of this story and the data provided by new techniques such as multiomics will allow us to better understand and manage this complex pathology.

## Figures and Tables

**Table 1 jcm-10-00389-t001:** Severity criteria according to ESCMID and IDSA/SHEA [16,38]: IDSA/SHEA criteria are shown on the grey background.

**Patient** **characteristics**	Age > 65 years old Immunosuppression	**Laboratory** **findings**	Leucocytes > 15,000 cells/mL Creatinine > 133 μmol/L
**Physical** **examination**	Fever > 38.5 °C Rigors Peritonitis Ileus Respiratory failure (mechanical ventilation) Hemodynamic instability		Left shift (bands > 20%) Albumine < 30 g/L Lactates > 5 mmol/L
		**Imaging**	Distension of large intestine > 6 cm Colonic wall thickening Pericolonic fat stranding Ascites (not explained by other causes)
**Endoscopy**	Pseudomembranous colitis		

**Table 2 jcm-10-00389-t002:** Cure rate and sustained clinical cure rate with metronidazole, vancomycin and fidaxomicin.

Study	TrT	Number of Patients	CC (%)	CC (%)			Rec (%)
			Global	Mild	Mod	Sev	
Johnson [49]	MTZ	278	72.7	78.7	73.9	66.3	23
	VCM	259	81.1 *	82.7	82.2	78.5 *	20.6 *
Teasley [50]	MTZ	42	88				2.7
	VCM	52	86				13.3
Wenisch [51]	MTZ	31	94				16
	VCM	31	94				16
Zar [52]	MTZ	79	84	90		76	14
	VCM	71	97	98		97 *	7
Louie [40]	VCM	309	85.8	85	83	88.6	25.3
	FDX	287	88.2	92.2	91.9	82.1	15.4 *
Cornely [39]	VCM	257	86.8	91.5		76.2	26.9
	FDX	252	87.7	91.7		73.4	12.7 *
Guery [48]	VCM	179	82				19
	EPFDX	177	78				6 *

Trt: treatment, CC: clinical cure, Rec: recurrence, Mod: moderate, Sev: severe, MTZ: metronidazole, VCM: vancomycin, FDX: fidaxomicin, EPFDX: extended-pulsed fidaxomicin. * *p* < 0.05 vs the control group.

## Data Availability

No data so excluded.

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
