# Peer review of "Clostridioides difficile Infection, Still a Long Way to Go"

_jcm, 2021, doi:10.3390/jcm10030389_

Round 1

Reviewer 1 Report

I read with a great interest this paper entitled "Clostridioides difficile infection: unmet needs and unsolved questions", which is very comprohensive review representing todays knowledge starting from diagnostic procedures, clinical signs, treatment possibilities and prevention of CDI. I think that such publication summaring whole knowledge about CDI is required now.

Authors not only mentioned current knowledge, but also asked questions to answer in the future studies.

I think that this publication will be of great interest of the physicians and microbiologists, especially as it poses openly of the questions that we often ask ourselves during routine diagnostic practice and during the treatment of CDI patients. 

I suggest to publish this paper in JCM.

Sincerely

Reviewer 2 Report

Row 42: Use the abbreviation CDI and omit C.difficile infection

Row 133-134 and 223: unclear sentence - please clarify! (...cover with high testing...?)

Rows 172,175,176: use EIAs

Rows 195,197: use CA

Row 308: extended administration - should be better clarified what it means

Table 1: last row - the abbreviation PDFDX should be opened in the footnote (and the meaning of it either here or in row 314)

Table 1: The column SCC ? - the percentages should be for cure, not recurrences

Row 332: start using FMT

Row 337: sentence wording

Row 382: Finnish

Row 404: unclear sentence - ...question of what?

Row 418: do you mean: long term follow-up after FMT?

Row 475: HA-CDI  - open?

Reviewer 3 Report

The authors present a very well-written, well-referenced review of CDI diagnosis, treatment and prevention. They cover the main controversies and offer a balanced, informed summary of the current status and recommendations.

My comments include only minor clarifications and suggestions.

Epidemiology section:
Lines 102-103:  I would say that NAAT assays for the detection of toxigenic C. difficile strains, rather than detection of ‘toxins’ to avoid confusion with direct toxin assays.

Lines 105-106: I would consider deleting this sentence or modifying it. The concern about missing cases using a NAAT assay that only targets tcdB is misplaced. Isolates that only encode the toxin A gene are exceedingly rare and are of uncertain significance. In contrast, there is a widely disseminated toxin A-negative, toxin B-positive strain (particularly in Asia) that has specific deletions in the toxin A gene and as a consequence produces no toxin A (Eichel-Streiber C, et al. FEMS Microbiol Letter 1999;178:163-8;  Kim H, et al. J Clin Microbiol 2008;46:1116-7 as examples).

Lines 219-220: I would agree with the authors that NAAT, as a standalone test is not an optimal diagnostic strategy because of the low clinical PPV in many studies. However, this has been a controversial recommendation in the U.S. as many clinical laboratories are reluctant to replace NAAT testing with an algorithm which includes toxin testing. Proponents of NAAT only testing will argue if there is careful screening of patients prior to submission of specimens and only if they only accept specimens from patients with diarrhea (>/= 3 unformed bowel movements) and no recent laxative use then the NAAT result is valid. The discussion in this section is very good and while I personally agree with this recommendation (NAAT as a standalone test is not appropriate), the authors might acknowledge the controversy across the pond.

Lines 221-224: I agree with this recommendation, but the authors might mention to avoid using EIA/LFA toxin assays that have particularly poor performance characteristics (i.e., some have very low sensitivity).

Treatment section:

Line 299: I would consider removing the clause ‘either alone or in association’.  I am not sure of the meaning intended here, but the only instance where dual therapy is recommended (by either ESCMID or IDSA/SHA) is in the situation of severe/complicated or fulminant disease where oral vancomycin and iv metronidazole is recommended and that recommendation is based on little evidence as pointed out by the authors.

Lines 336-338: I would agree with the authors that a standard course of fidaxomicin given to patients with multiple CDI recurrences is likely not as effective as when given to patients with a primary CDI episode or a first recurrent CDI episode that a pulsed vancomycin or FMT is typically recommended in this setting. An extended, pulsed fidaxomicin regimen might be considered as well although there is only anecdotal evidence supporting this approach (Soriano MM, et al  Open Forum Infect Dis 2014 Aug 25;1(2):ofu069). In addition, the last word in this sentence ‘compared’ might be replaced with ‘considered’.

Lines 393-395: The reference for this statement (increase of cure rate with multiple FMT infusions) is missing. I believe the correct reference should be their reference #76 (Lee CH, et al. JAMA 2016;315:142-9).

Lines 407: I would clarify ‘diversity’ to say ‘microbial diversity of the colonic microbiome’ could be a potential end point.

Line 428: I agree with the authors that a better definition of severe infection with validated criteria is needed, but if metronidazole is no longer recommended, I am not sure if this definition is as clinically important (just a personal comment, no need to change summary).

Line 433-434: I am in total agreement with this point. Data comparing extended-pulsed fidaxomicin to standard 10-day treatment course of fidaxomicin and comparing to pulsed vancomycin is needed!

Line 437: I also agree that this question of using FMT after the first CDI recurrence is worth exploring, but only if there is some assurance of the safety of the FMT product (i.e., regulatory approval) otherwise, the risk/benefit needs to be taken into account. While the risk of a subsequent recurrence after a first CDI episode is higher than after a first episode, most patients can be managed with appropriate antibiotics at this point.

Line 456: This sentence should be ‘colonization is more frequent THAN active disease..’

Lines 466-467: What are the authors proposing as a ‘competition against colonization by toxigenic strains’? Are they referring to the potential of non-toxigenic strains of C difficile to be protective against toxigenic strains (Gerding DN et al. JAMA 2015;313:1719-27)?
